# Recovery of Gold from the Refractory Gold Concentrate Using Microwave Assisted Leaching

**Kanghee Cho [1], Hyunsoo Kim [2], Eunji Myung [2], Oyunbileg Purev [2], Nagchoul Choi [1,\*] and Cheonyoung Park [2,\*]**

[1] Research Institute of Agriculture and Life Sciences, Seoul National University, Seoul 08826, Korea; kanghee1226@hanmail.net

[2] Department of Energy and Resource Engineering, Chosun University, Gwangju 61452, Korea; star8538@naver.com (H.K.); ej6865@naver.com (E.M.); oyunbileg@chosun.kr (O.P.)

\* Correspondence: nagchoul@empas.com (N.C.); cybpark@chosun.ac.kr (C.P.); Tel.: +82-62-230-7878 (N.C.)

**Abstract:** Microwave technology has been confirmed to be suitable for use in a wide range of mineral leaching processes. Compared to conventional leaching, microwave-assisted leaching has significant advantages. It is a proven process, because of its short processing time and reduced energy. The purpose of this study was to enhance the gold content in a refractory gold concentrate using microwave-assisted leaching. The leaching efficiencies of metal ions (As, Cu, Zn, Fe, and Pb) and recovery of gold from refractory gold concentrate were investigated via nitric acid leaching followed by microwave treatment. As the acid concentration increased, metal ion leaching increased. In the refractory gold concentrate leaching experiments, nitric acid leaching at high temperatures could limit the decomposition of sulfide minerals, because of the passive layer in the refractory gold concentrate. Microwave-assisted leaching experiments for gold recovery were conducted for the refractory gold concentrate. More extreme reaction conditions (nitric acid concentration > 1.0 M) facilitated the decomposition of passivation species derived from metal ion dissolution and the liberation of gangue minerals on the sulfide surface. The recovery rate of gold in the leach residue was improved with microwave-assisted leaching, with a gold recovery of ~132.55 g/t after 20 min of the leaching experiment (2.0 M nitric acid), according to fire assays.

**Keywords:** recovery; gold; refractory; nitric acid; microwave

## 1. Introduction

Gold has excellent physical and chemical properties, and is one of the most important noble metals. The current rapid decline in high-grade gold ores and readily available low-grade gold ores has made the mineral processing industry increasingly reliant on complex and refractory gold ores [1,2]. Consequently, mineral processing challenges related to the complexities of ore mineralogy and the process parameters, such as the impacts of associated minerals, are important research questions [3]. Mineralogical studies aim to characterize complex sulfides and show the interrelations of the target minerals in the refractory minerals. In addition, these studies usually analyze gold-bearing sulfide minerals for follow-up processes and efficient gold recovery. Gold can be found in complex sulfide minerals, not only due to the presence of invisible gold, but also due to the existence of solid solutions [4]. Gold is commonly associated with sulfide minerals, particularly pyrite and arsenopyrite [5]. A refractory gold ore typically contains different sulfide minerals with various gold concentrations [6]. Gold is highly encapsulated in the sulfide matrix in refractory gold ores. Pretreatment is an important process to recover gold from sulfide minerals. Gold-bearing ores ordinarily contain complex or refractory ores of sulfide minerals that interfere with gold recovery efficiency. It is necessary to remove sulfide minerals

prior to leaching, and the most appropriate pretreatment is flotation and roasting. However, most of the sulfide minerals in a refractory gold concentrate react to form a harmful gas during roasting and the iron oxide produced during roasting encapsulates invisible or fine gold [7]. Pyrite is the most abundant sulfide mineral in refractory gold ore and its oxidation is thermodynamically favorable in the environment [8].

The surface oxidation of pyrite is a particularly important step for leaching valuable metals from sulfide minerals. The presence of invisible gold has been established in pyrite, which is a common type of gold-bearing sulfide mineral that is mostly found in association with other sulfide minerals. The gold present in sulfide minerals can be divided into visible gold and invisible gold. While visible gold can be observed with an optical microscope, invisible gold is very difficult to observe with these microscopes. The size of invisible gold may be on the order of nanometers, its presence is indicated by significant refractory imposed by the sulfide minerals (e.g., arsenopyrite, pyrite) [9]. The hindrance of gold recovery from complex sulfide minerals has been attributed to the formation of a passive layer on the mineral surface. Recent studies [10–14] have been conducted on the passive layer effect in sulfide minerals and the role played by sulfur in leaching solutions. In particular, the leaching of galena (PbS) has some problems, due to the low solubility of galena without the presence of an oxidant, the formation of precipitation, and the disposal of a large amount of iron that dissolves along with the lead. Therefore, leaching of the galena has been investigated by many researchers, who have proposed different nitric acid based leaching systems, including $HNO_3$ [10], $H_2O_2$-$HNO_3$ [13] and Fe(III)-$HNO_3$ [14]. The kinetics of lead dissolution from galena are slow, due to the formation of the passivation layer on the surface of sulfide by oxidation of sulfide, to form elemental sulfur under oxidative conditions. It is known that a partial dissolution or decomposition of sulfide minerals by the leaching solution lowers the leaching rate by forming a passive layer [11].

Unfortunately, the contributions of reactive sulfur species and the mechanisms of interaction with the leach residue surface are unclear. Therefore, studying the effects of relational minerals on gold is important. The processing of intractable substances presents challenges related to their complexity or refractory minerals. Increasing concerns regarding environmental protection has triggered efforts to identify alternatives that are environmentally friendly. Due to environmental concerns over cyanidation in hydrometallurgy, considerable attention has been dedicated to alternative non-cyanide solutions for refractory gold mineral leaching. Nitric acid has been recognized as one of the most promising reagents to pre-treatment process, and reduce cyanide consumption by refractory gold concentrates. The nitric acid leaching process is advantageous in that it can produce highly oxidizing conditions and is therefore an effective leaching agent for most sulfides [10]. However, a disadvantage of the nitric acid leaching process is the oxidation of sulfide, both to elemental sulfur and sulfate, in many cases in almost equal parts. This results in an increase in reagent consumption and the necessity of handling sulfate [10,13].

The use of microwave-assisted leaching processes has several advantages, including reduced energy consumption and elevated reproducibility [15,16]. These characteristics are the main drivers of metal ions in complex sulfide minerals. For example, selective leaching was successfully applied to lead smelting residues. Kim et al. (2017) reported that, when microwave assisted extraction (MAE) and autoclave leaching were performed to solubilize other valuable metals (Cu, Ni, Zn) from the matte, MAE has higher oxidation power than autoclave leaching [11]. This can explain the higher conversion of Fe sulfides to Fe oxides compared to autoclave leaching, as well as the higher leaching efficiency of Cu, Ni and Zn from their sulfides. Moreover, MAE is a simple process that can save energy and processing costs. Microwave heating has also been applied by Choi et al. [17] to perform pre-treatment, followed by a thiourea leaching step for gold extraction from gold concentrate. Due to the many advantages of microwaves, they have been widely used in mineral pretreatment. The main purpose of this work is to investigate the leaching behavior of the metals As, Cu, Fe, Zn, and Pb and passive layer decomposition in the refractory gold concentrate associated with nitric acid leaching under various

conditions, using a microwave system. Furthermore, to increase the recovery of gold, nitric acid was used during microwave-assisted leaching.

## 2. Materials and Methods

### 2.1. Refractory Gold ore and Concentrates

The refractory gold ore and concentrate were obtained from a gold mine in Haenam, Korea. Refractory gold ores, including sulfide minerals such as pyrite and chalcopyrite, were used to investigate the influence of mineralogical characteristics on nitric acid leaching. A polished section of the ore mineral was prepared and studied microscopically under reflected light, to identify mineralogical properties. Polished sections were prepared by placing refractory gold ore in an epoxy resin which, after curing, was polished to ensure flatness. The textures of pyrite in the gold ore were investigated using nitric acid etching. The etching method involves the application of a few drops of 65% nitric acid on the polished mounts.

The refractory gold concentrate was obtained through a flotation process. The mineral composition of the refractory gold concentrate was analyzed using XRD. The chemical characterization of the surface species was performed using X-ray photoelectron spectroscopy (XPS). Gold is an element with a severe nugget effect [7,18], which may cause errors in the analysis based on the sampling process. To minimize this, the whole sample was sufficiently mixed and the sample was prepared using the cone and quartering method. To determine the chemical composition of the refractory gold concentrate, the sample was digested with aqua regia. The solution chemistry was analyzed using ICP-OES. The content of gold was analyzed using a fire assay [19,20].

### 2.2. Leaching Experiment

#### 2.2.1. Microwave-Assisted Leaching: Effect of Nitric Acid Concentration and Temperature

The samples were sieved through a < 170 mesh screen. For the leaching experiment of the refractory gold concentrate, a microwave system (2.45 GHz, MARS 6, CEM Corporation, Matthews, NC, USA) was used. The microwave system was equipped with a digital temperature control sensor, which allowed the temperature to be accurately measured in real-time. The leaching experiments were conducted at different temperatures (40, 80, and 120 °C) and nitric acid concentrations (0.1, 0.5, and 1.0 M). Each leaching experiment consisted of several heating steps to reach a set temperature. A 100 mL disposable Teflon vessel containing 20 mL of $HNO_3$ at various concentrations and 1.0 g of the refractory gold concentrate was heated in the microwave. At the end of the reaction, the microwave system was switched off and the reactor was allowed to cool to room temperature. The oxidation and reduction potentials of the leaching solution were measured using an ORP (Orion 3-Star, Thermo Fisher Scientific, Santa Clara, CL, USA) meter. The leaching solution was filtered through a syringe filter. The weight of the leach residue was measured after leaching. The solution was analyzed by ICP-OES. Each experiment was performed in duplicate. The amount of metal ions leached was calculated using the following equation:

$$L = \frac{M_1}{M_0} \times 100 \tag{1}$$

where L is the leaching percentage of metal ions, $M_0$ is the metal ion content of the sample before leaching, and $M_1$ is the metal ion content of the sample after leaching.

The rate of metal ion leaching was determined using the following equation [17]:

$$E = E_I\left(1 - e^{-kt}\right) \tag{2}$$

where E (%) is the metal ion concentration in the leaching solution at time t, $E_I$ (%) is the maximum concentration of metal ions, and $K$ ($min^{-1}$) is the leaching rate constant.

2.2.2. Microwave-Assisted Leaching: Recovery of Gold

Microwave-assisted leaching was performed by heating a 500 mL Pyrex glass containing 200 mL of nitric acid (1.0, 3.0, and 5.0 M) and 20 g of the refractory gold concentrate. Figure 1 shows the schematic of the laboratory-scale microwave-assisted leaching developed in this study. The flask was placed in the center of the microwave oven and heated. The processing time was set to 15 min for all experiments. After the reaction period was over, the contents were cooled to ambient temperature and removed for solid-liquid separation. After each leaching experiment, the leach residue was filtered through filter paper and the content of gold in the leach residue was analyzed by fire assay. The weight of the leach residue was measured after leaching.

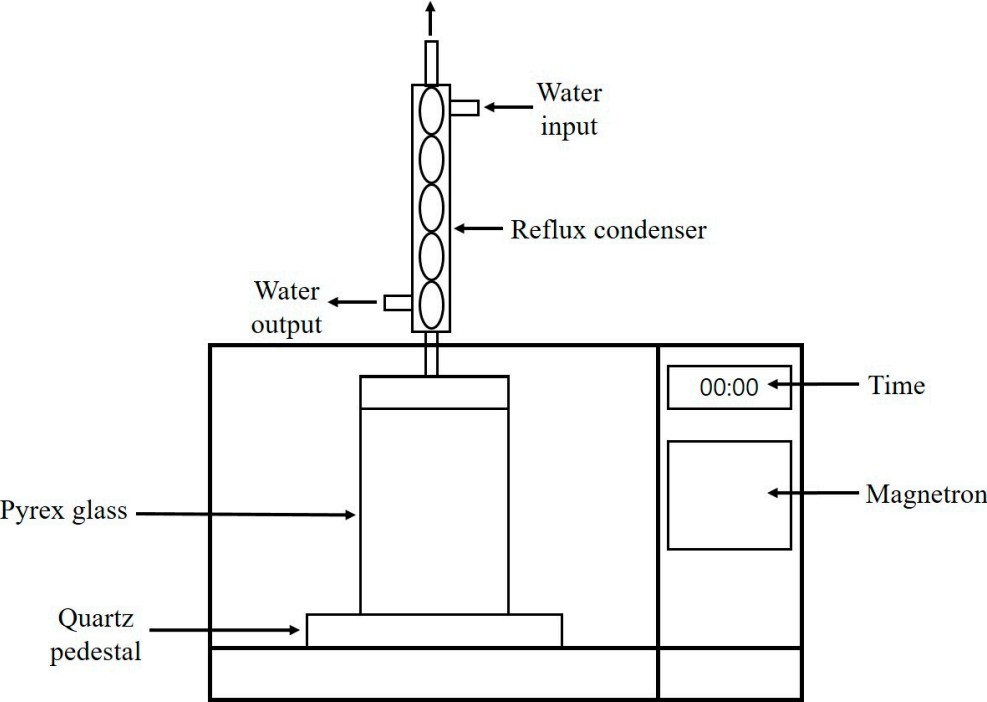

**Figure 1.** Schematic diagram of microwave-assisted leaching system.

*2.3. Characteristics of the Residue*

The leach residues obtained were analyzed by XPS and XRD to determine the surface arsenic, sulfur, and iron species and the mineralogical composition, respectively. Morphological studies were carried out on the leach residue using SEM-EDS (Scanning Electron Microscopy-Energy-dispersive X-ray spectroscopy) (S4800, Hitachi, Matsuda, Japan).

*2.4. Analysis Method*

The particle specific surface area of the leach residue was determined by laser diffraction (Mastersizer 2000, Malvern Panalytical Ltd., Malvern, UK). After the leaching experiment, the leaching solution was analyzed using inductively coupled plasma optical emission spectrometry (ICP-OES; Optima Model 5300 DV, Perkin Elmer, Norwalk, CT, USA), to determine the amount of metal leaching. The chemical characterization of the surface species was performed by XPS (K-Alpha+, Thermo Fisher Scientific, Waltham, MA, USA) to determine As, Fe, and S elements in the measured sample. The refractory gold concentrate and leach residue were subjected to XRD (X'Pert Pro MRD, PANalytical, Almelo, Netherlands) analysis. Cu-K$\alpha$ X-rays were used and 2$\theta$ of 10°–70° was analyzed with an acceleration voltage of 40 kV, a current of 30 mA, and a scanning speed of 2°/min.

## 3. Results

### 3.1. Characteristics of Refractory Gold Ore and Concentrate

The refractory gold mineral sample was observed using an optical microscope. As described in Figure 2, the internal textures of the pyrite samples were investigated using nitric acid etching. Minerals containing arsenic are deeply etched by nitric acid; therefore, this procedure can greatly assist in highlighting pyrite growth zones [18]. However, arsenopyrite was not found. Optical microscopy images of pyrite with core and rim zones were mainly targeted for this study. The main features observed were pyrite and chalcopyrite; native gold and electrum were not found. The general crystals were angular and irregularly shaped, with a grain size distribution. Some typical textures, such as pores and cracks, were found in the pyrite. Chalcopyrite crystals contained in the pyrite were also observed.

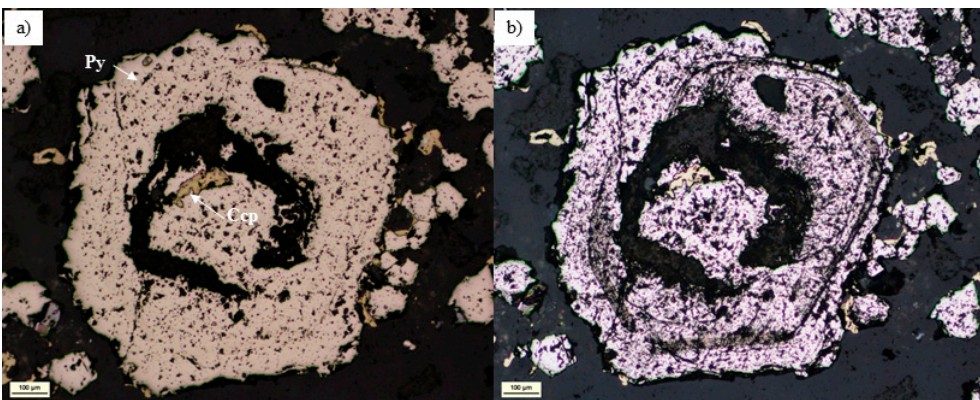

**Figure 2.** Microphotographs of gold ore samples. Reflected light microscopy of the etching of pyrite grains (**a**) before and (**b**) after etching in $HNO_3$ solution. Ccp is chalcopyrite and Py is pyrite.

Figure 3 presents the SEM-EDS analysis for the refractory gold concentrate. Idiomorphic cubic pyrite was observed. Generally, the Fe:S ratio is approximately 1:2 in pyrite; however, this phase has an S/Fe ratio of > 3, identical to that of pyrite. EDS analysis of the surface of pyrite showed the presence of arsenic, iron, sulfur, and gold, suggesting the formation of sulfur-containing arsenic and iron compounds. XRD analysis of the refractory gold concentrate revealed that consists of pyrite and quartz. The main chemical composition of the concentrate is listed in Table 1.

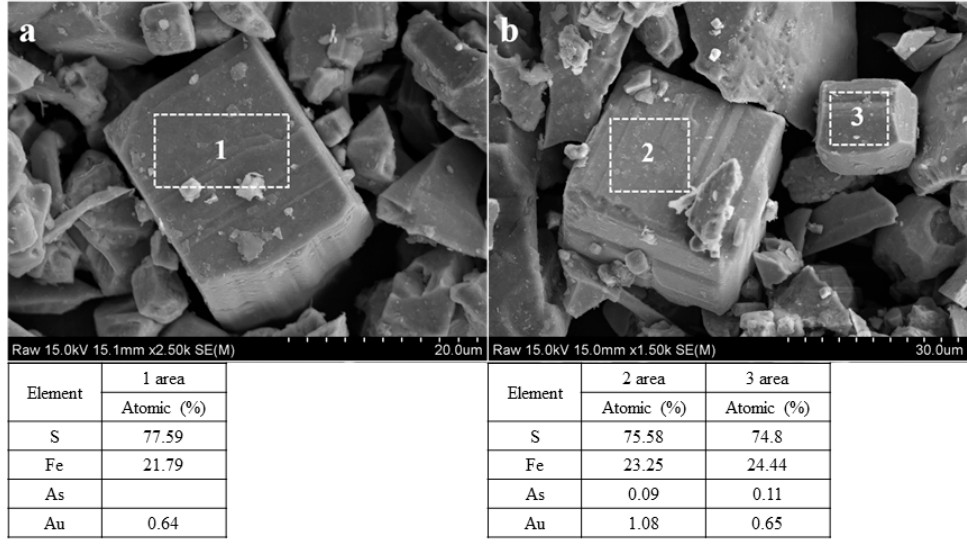

| Element | 1 area |
| --- | --- |
|  | Atomic (%) |
| S | 77.59 |
| Fe | 21.79 |
| As |  |
| Au | 0.64 |

| Element | 2 area | 3 area |
| --- | --- | --- |
|  | Atomic (%) | Atomic (%) |
| S | 75.58 | 74.8 |
| Fe | 23.25 | 24.44 |
| As | 0.09 | 0.11 |
| Au | 1.08 | 0.65 |

**Figure 3.** SEM-EDS (Scanning Electron Microscopy-Energy-dispersive X-ray spectroscopy) of pyrite in the refractory gold concentrate.

**Table 1.** Chemical composition of the complex sulfide concentrate.

| Cu (wt%) | Pb (wt%) | Zn (wt%) | Fe (wt%) | As (g/t) | Au (g/t) |
|----------|----------|----------|----------|----------|----------|
| 0.19 | 0.15 | 0.63 | 43.86 | 0.10 | 94.37 |

## 3.2. Leaching Experiments

### 3.2.1. Effect of Nitric Acid Concentration

As shown in Figure 4, As, Cu, Fe, and Zn show increasing leaching efficiency with increasing nitric acid concentration from 0.1 to 1.0 M. Au was not detected under most of the experimental conditions. The increase in the concentration of nitric acid should accelerate the reaction rate of the refractory gold concentrate. As this effect is evident only at the highest nitric acid concentrations, it suggests that oxidation is the most likely factor. Based on the experimental results, the leaching parameters of the leaching experiment conditions estimated using Equation (2) are presented in Table S1. Initially, the metal ion leaching rate was slow, but steadily increased with increased nitric acid concentration. The sample mass decreased from 19.0% at 0.1 M, 27.0% at 0.5 M and 51.5% at 1.0 M. Nitric acid concentration was shown to be effective for the decomposition of the dominant sulfide minerals, due to the oxidation of insoluble sulfides to water-soluble sulfate phases during leaching.

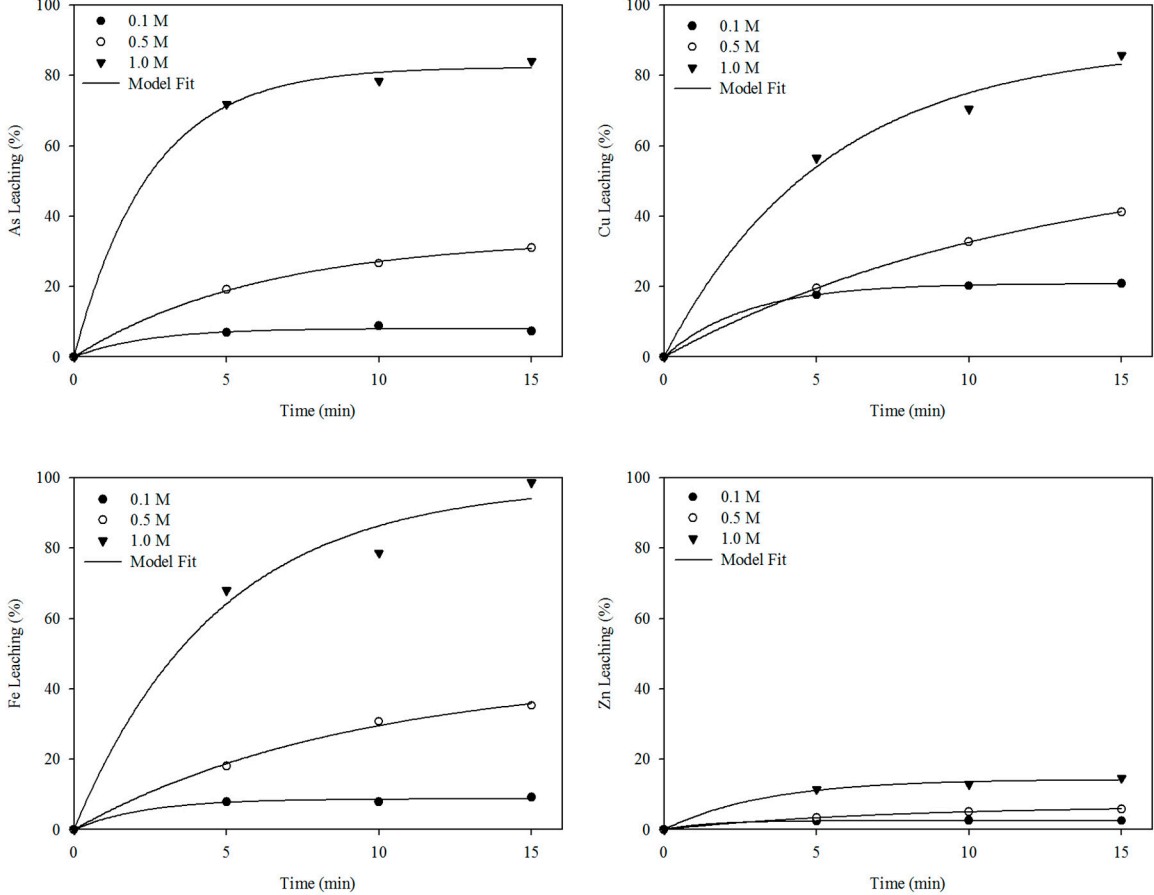

**Figure 4.** Effect of $HNO_3$ concentration on the leaching efficiencies of As, Cu, Zn, and Fe from the refractory gold concentrate. The leaching conditions were a reaction time of 15 min, reaction temperature of 80 °C, and $HNO_3$ concentration of 0.1, 0.5, or 1.0 M. According to Equation (2) in Table 2.

**Table 2.** Refractory gold concentrate and leach residue surface atomic composition with different treatments. The leaching conditions were a reaction temperature of 80 °C reaction time of 15 min, and HNO₃ concentration of 0.1 and 0.5 M.

| Experiment | S2p (%) | Fe2p (%) | S/Fe ratio (%) | As3d (%) |
|---|---|---|---|---|
| Raw | 44.52 | 39.64 | 1.12 | 6.39 |
| Residue 0.1 M | 77.79 | 17.93 | 4.34 | 0.11 |
| Residue 0.5 M | 78.50 | 18.69 | 4.20 | - |

### 3.2.2. Effect of Temperature

The results are presented in Figure 5. The As leaching increased from 84.0% to 92.5%, as the temperature increased from 80 to 120 °C. Initially, the As leaching rate was slow, but steadily increased with increased temperature. However, Cu leaching decreased from 85.8% at 80 °C to 75.8% at 120 °C and Zn leaching decreased from 14.6% at 80 °C to 10.5% at 120 °C. The Fe leach ability was 98.7% at 80 °C and slightly decreased to 94.6% at 120 °C. As the temperature increased from 80 to 120 °C, the Cu leaching rate constant increased from 0.18 to 0.36 min$^{-1}$ and the Fe leaching rate constant increased from 0.21 to 0.66 min$^{-1}$. Based on the experimental results, the leaching parameters of the leaching experiment conditions estimated using Equation (2) are presented in Table S2. Due to the effect of temperature, it is necessary for sulfide to be oxidized into sulfate, with a minimized formation of elemental sulfur that impedes the subsequent recovery of gold.

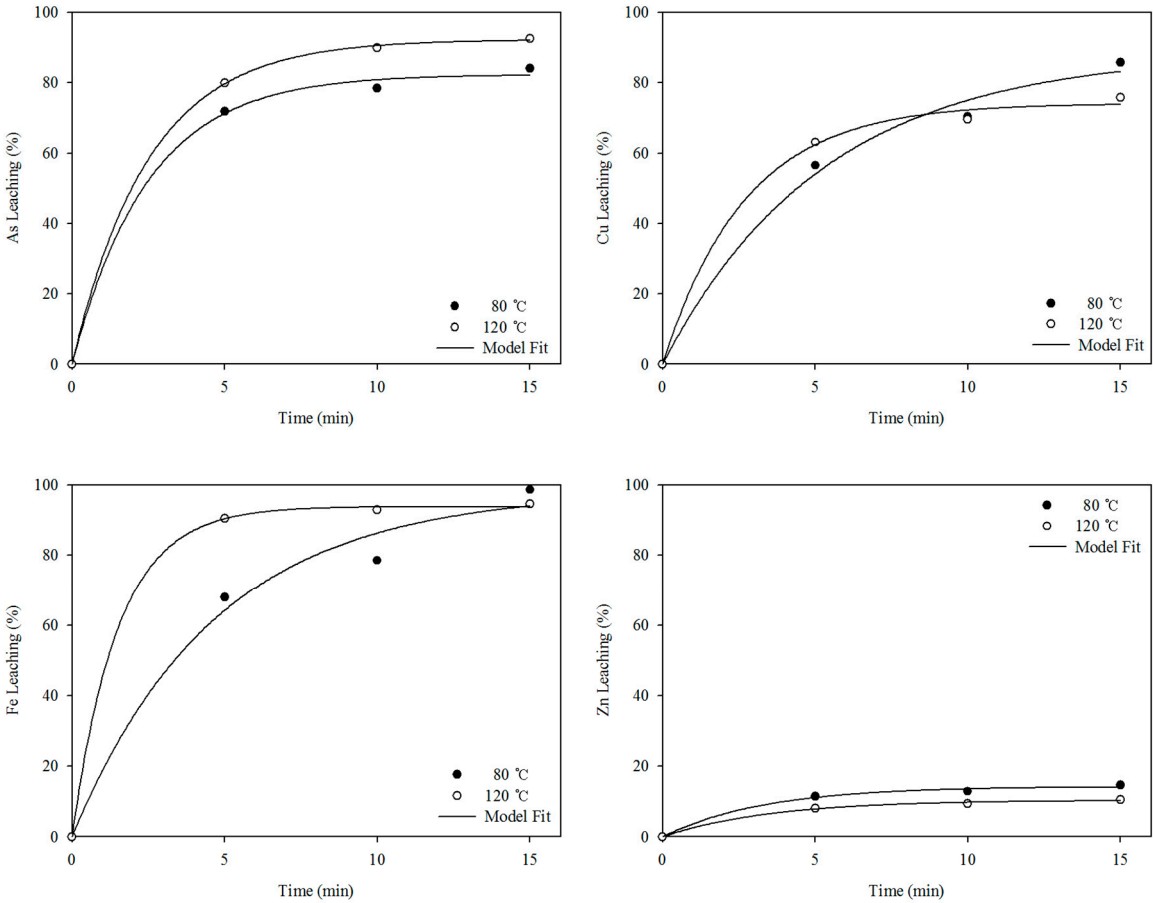

**Figure 5.** Effect of temperature on the leaching efficiencies of As, Cu, Zn, and Fe from the refractory gold concentrate. The leaching conditions were a reaction temperature between 80 and 120 °C, HNO₃ concentration of 1.0 M, and reaction time of 15 min. According to Equation (2) in Table S2.

Figure 6 shows the effect of temperature on the Pb leach ability from the refractory gold concentrate at nitric acid 1.0 M. Pb leaching was increased for the first 5 min; the rate decreased after that, and leaching efficiency decreased from 57.8% at 40 °C, 15.3% at 80 °C, and was not detected at 120 °C. The ORP (mV) was reduced by leaching, particularly at the end of the leaching time. The ORP slowly decreased to 686 mV at 40 °C, 638 mV at 80 °C, and 623 mV at 120 °C (Figure 6b). These results indicate that lead complexes, such as $PbNO^{3+}$, $Pb(NO_3)_2$, $Pb(NO_3)_3^-$ and $Pb(NO_3)_4^{2-}$, are formed by oxidation at high-temperatures [10]. The respective reactions may involve the formation of a passive layer of elemental sulfur, which is then leached, and passivation by lead sulfate or basic sulfate [10]. The lower Pb leaching at > 80 °C is likely due to passivation by increased elemental sulfur formation. This shows that it is possible to obtain lead in solution, although lead may not precipitate at lower potentials. XRD analysis of the leach residues mainly show untreated pyrite, with a small quantity of quartz and sulfur. The reaction produced elemental sulfur (Figure 7) and there is a clear indication that surface passivation prevented further leaching. Therefore, the decrease in Pb leaching efficiency may be due to the consumption of oxidants K and acids.

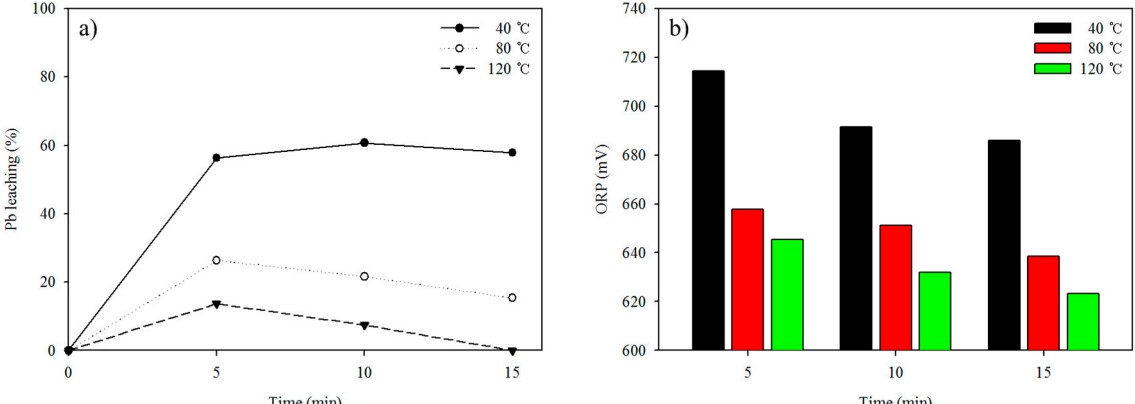

**Figure 6.** Effect of temperature on the leaching efficiencies of Pb (**a**) and ORP (mV) (**b**) from the refractory gold concentrate. The leaching conditions were an $HNO_3$ concentration of 1.0 M, reaction time of 15 min, and reaction temperatures of 40, 80, and 120 °C.

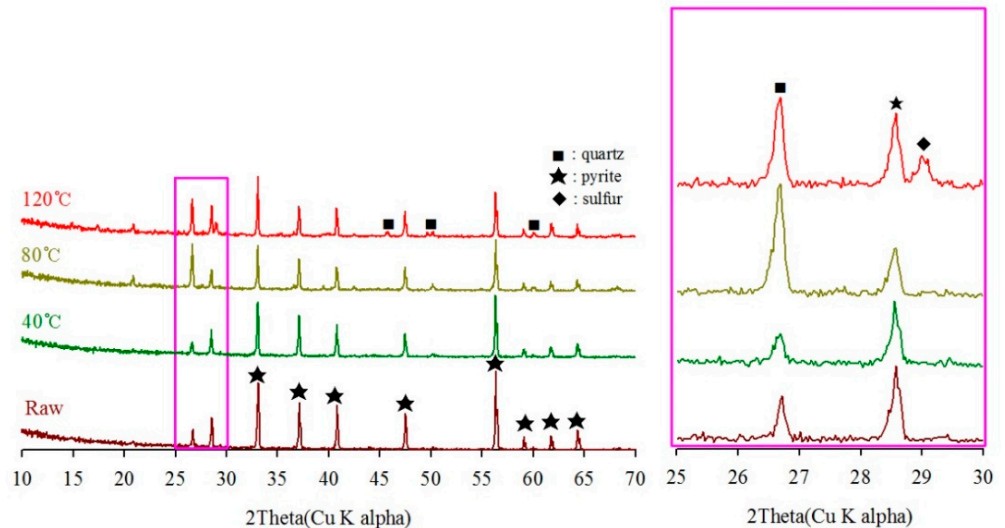

**Figure 7.** XRD patterns of raw and leach residue from nitric acid leaching at 40, 80, and 120 °C. The leaching conditions were an $HNO_3$ concentration of 1.0 M and a reaction time of 15 min.

As the temperature increased, Pb leaching decreased, which mainly occurred due to the formation of a passive layer. The leaching efficiencies of Fe in the leaching experiment were not significant

(Figure 8). The Fe leaching rate was 90.5% at 40 °C, 93.1% at 80 °C, and 94.61% at 120 °C after 15 min. Sulfide oxidation [21] is strongly affected by temperature (> 100 °C) and a temperature increase has a negative effect on Pb leaching. It is noteworthy that the Pb leaching efficiency was lower than the Fe leaching efficiency. Therefore, it is difficult to simultaneously leach Pb and other base metals, such as As, Cu, and Zn, from sulfide minerals in nitric acid at elevated temperatures. In addition, Pb shows a different leaching behavior compared with the other metal ions, due to the leaching of elemental sulfur to sulfate that occurs after the complete oxidation of sulfide to sulfur [13].

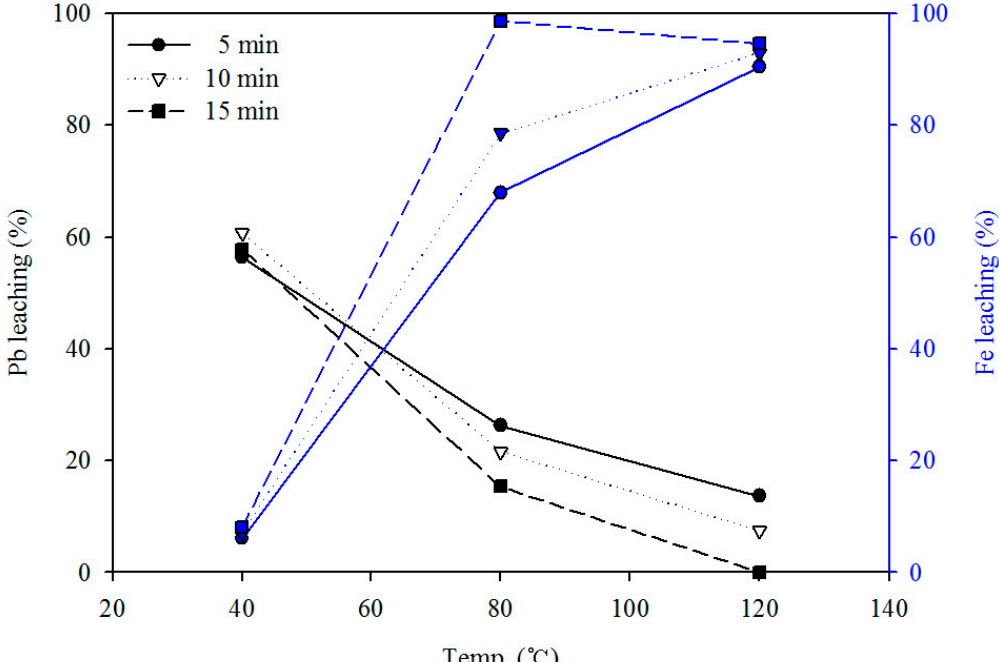

**Figure 8.** Effect of temperature on the leaching efficiencies of Pb and Fe from the refractory gold concentrate. The leaching conditions were an $HNO_3$ concentration of 1.0 M and a reaction temperature of 40, 80 and 120 °C.

### 3.3. Characterization of the Passive Layer in the Refractory Gold Minerals

XPS analysis was conducted to detect the dissolution changes of metal ions (S, Fe, and As) on the surface of minerals. Table 2 shows the changes in the chemical environment of S, Fe, and As, before and after leaching. These results indicate a change in the surface of the metal ions during nitric acid leaching. The Fe atom percent was 39.64% in the raw sample and 17.93% at 0.1 M, 18.69% at 0.5 M in the nitric acid leaching residue, indicating that the superficial pyrite in the minerals did not significantly dissolve during the reaction time of 10 min. Some Fe existed in the crystal lattice of the sulfide minerals in the refractory gold concentrate, due to the superficial pyrite on the minerals dissolving. The XPS analysis indicates that the pyrite in the refractory gold concentrate did not significantly dissolve during the nitric acid leaching at < 0.5 M. The S/Fe ratio determines the surface of sulfide mineral decomposition and affects the efficiency of passive layer removal [22]. S/Fe ratio in nitric acid leaching (reaction time of 10 min) reaches 4.34 at 0.1 M and 4.20 at 0.5 M. This indicates that a relatively lower proportion of iron was consumed. However, the atomic percentages of arsenic on the surface of the raw concentrate decreased from 6.39% to 0.11% at 0.1 M and was not detected at 0.5 M. This indicates that a relatively large proportion of arsenic was consumed.

According to the XPS reference [23], the Fe2p peak with a binding energy of 708.17 eV and the S 2p peak with a binding energy of 160.93 eV belong to pyrite. In the nitric acid leaching residue from leaching at < 0.5 M, the Fe2p peak with a binding energy of 708.54 eV and the S2p peak with a binding energy of 161.02 eV were assigned to pyrite, both having small changes compared with

those in the raw sample. The S2p spectrum demonstrates that S exists in multiple oxidation states, including monosulfide ($S^{2-}$), disulfide ($S_2^{2-}$), polysulfide ($S_n^{2-}$), elemental sulfur ($S^0$), thiosulfate ($S_2O_3^{2-}$), and sulfate ($SO_4^{2-}$). Raw sample disulfide species of sulfide minerals are oxidized to sulfate via several intermediate steps [23,24]. The leaching of sulfide is complex, involving the dissolution of various sulfur species. For the leaching of sulfide minerals, the preferential release of metal ions such as Fe, Cu, Zn, etc., into the leaching solution results in the passive layer of surface $S^{2-}$, accounting for the formation of $S_2O_3^{2-}$ and $SO_4^{2-}$. This was possibly due to the S-S bonding being weaker than the Fe-S bonding, meaning that the S-S bonds were more easily broken in the lattice pyrite. The passivation layer on the residue surface mainly consisted of iron and sulfur species, as seen in Figures 9 and 10.

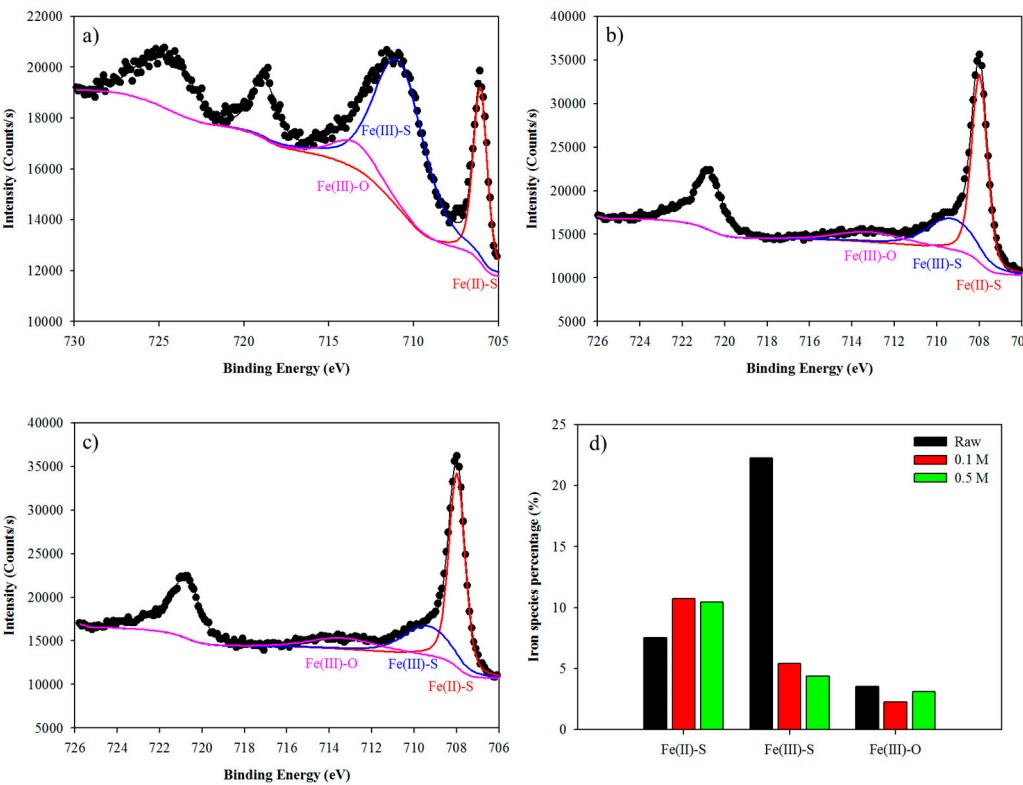

**Figure 9.** XPS spectra of Fe2p for (**a**) raw concentrate, (**b**) leach residue (0.1 M), (**c**) leach residue (0.5 M), and (**d**) distribution of the surface Fe species. The leaching conditions were 80 °C and 15 min.

Under different conditions, refractory gold concentrate leaching results in the formation of a passive layer on its surface, consisting of sulfur- and iron-rich layers. XRD analyses were conducted to confirm the changes in refractory sulfide minerals during leaching; the results of these analyses are presented in Figure 11. The intensities of the pyrite peak (111) decreased, while those of the gangue peak increased in the leach residues. Sulfur was not found in the leach residues, which means that the sulfur in the sulfides was transformed into thiosulfate or sulfate, as opposed to elemental sulfur. The (111) plane shows a higher oxidation rate than the (100) and (110) planes [24]. This is likely due to the S-S bond in the pyrite, which is weaker than that in Fe-S. This indicates that the fraction of dissolved metal ions increased as the concentration of nitric acid increased. The SEM image (Figure 12) shows the morphology of the residue after leaching for 10 min, which is not noticeably different from that of the refractory gold concentrate. However, the surface of the leach residue became uneven; it is obvious that multiple holes appeared on the surface of the leach residue particles. It can be concluded that pyrite in the sulfide minerals is not decomposed by nitric acid at < 0.5 M (Figure 11). Therefore, we investigated the increase in nitric acid concentration to enhance the recovery of gold.

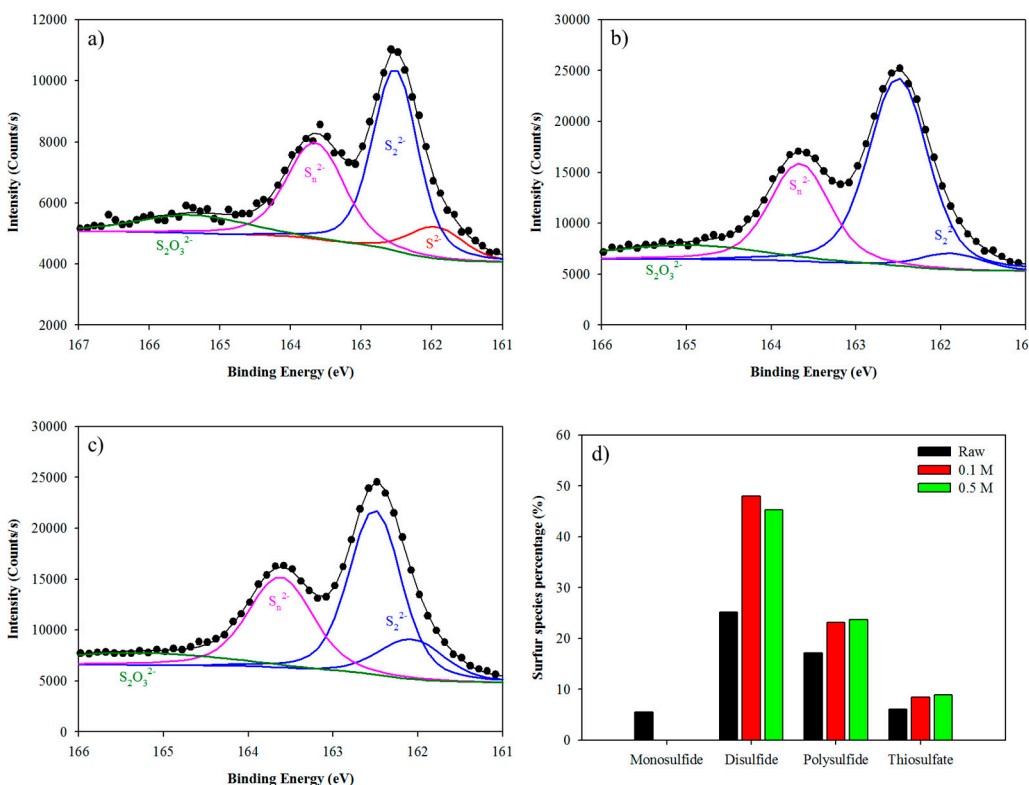

**Figure 10.** XPS spectra of S2p for (**a**) raw concentrate, (**b**) leach residue (0.1 M), (**c**) leach residue (0.5 M), and (**d**) distribution of the surface S species. The leaching conditions were 80 °C and 15 min.

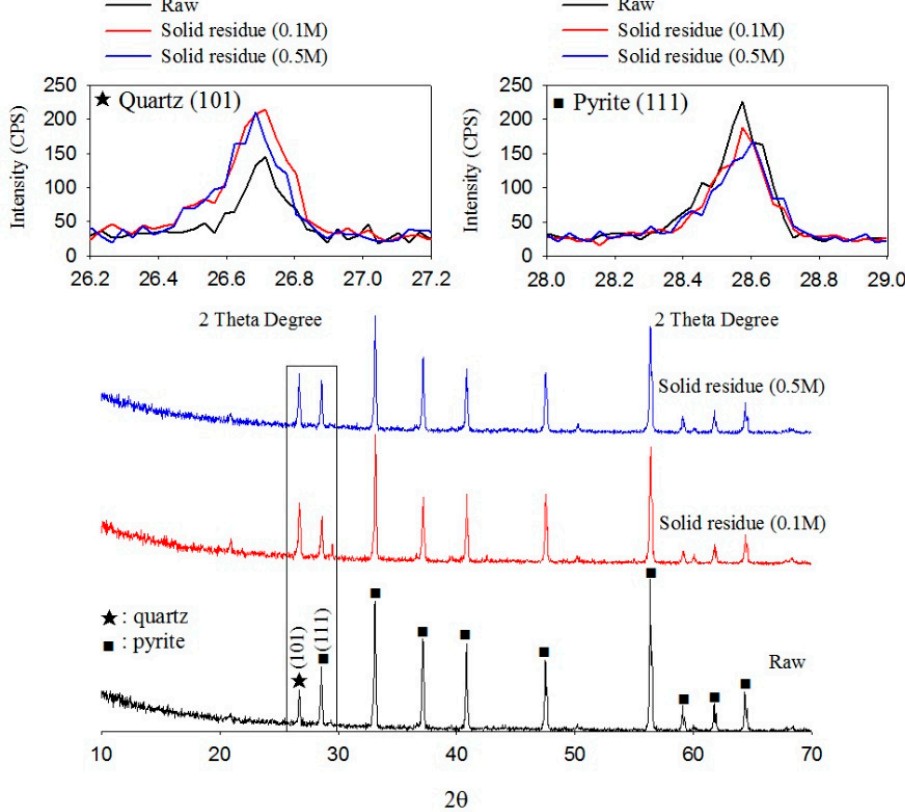

**Figure 11.** XRD patterns of raw and leach residue from nitric acid leaching at 0.1 and 0.5 M. The leaching conditions were 80 °C and 15 min.

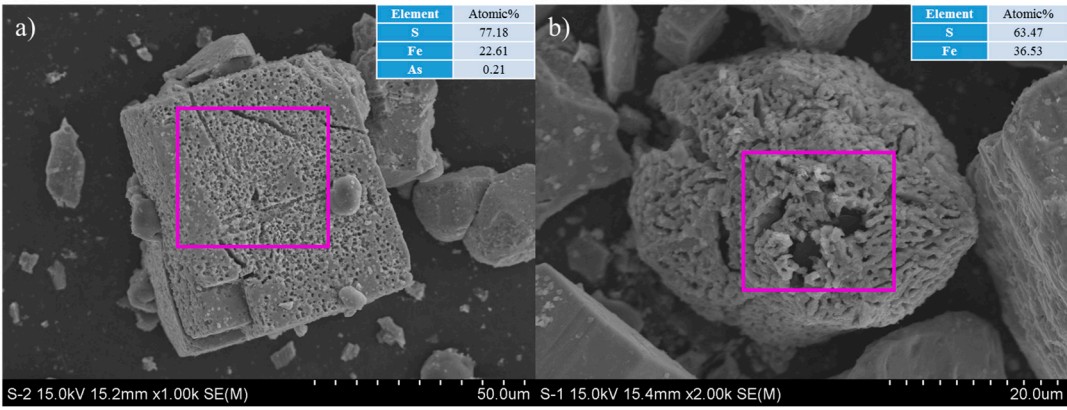

**Figure 12.** SEM images of leach residues. The leaching conditions were 80 °C, 15 min, and HNO$_3$ concentration of (**a**) 0.1 M and (**b**) 0.5 M.

### 3.4. Recovery of Gold by Microwave-Assisted Leaching

The gold content of the leach residue was analyzed using a fire assay. The effect of nitric acid concentration on gold recovery was studied, with a leaching time of 15 min (Figure 13). The original sample showed a gold content of 94.37 g/t according to the fire assay. After the leaching process with 3.0 M nitric acid for 10 min, a gold content of 126.39 g/t was obtained. Compared to the untreated refractory gold concentrate, the higher gold recovery confirms that a large portion of the gold was refractory in nature, with the gold occurring as either solid solution components in sulfide minerals or encapsulated in the sulfide minerals. The weight of the leach residue decreased when the nitric acid concentration increased. The sample mass decreased by a maximum of 69.60% after leaching at 5.0 M, indicating that the sulfide minerals were decomposed or dissolved by microwave-assisted leaching.

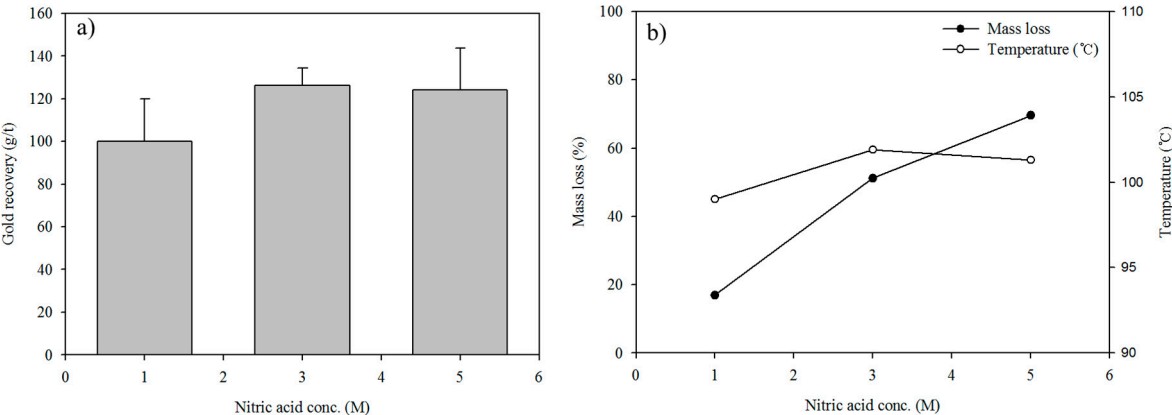

**Figure 13.** Effect of HNO$_3$ concentration on the (**a**) recovery of gold and (**b**) mass loss and reaction temperature from the refractory gold concentrate. The leaching conditions were a reaction time of 15 min and an HNO$_3$ concentration of 1.0, 3.0 and 5.0 M.

Microwave-assisted leaching experiments for gold recovery were conducted for the refractory gold concentrate for different reaction times. The results showed that the reaction time enhanced the recovery of gold. The gold recovery was approximately 132.55 g/t after 20 min of leaching with 2.0 M nitric acid, whereas the mass loss in leach residue similarly decreased with reaction time (Figure 14). The sample mass decreased by a maximum of 40.5% after 20 min. The microwave-assisted leaching experiment resulted in the loss of the constituents in the refractory concentrate.

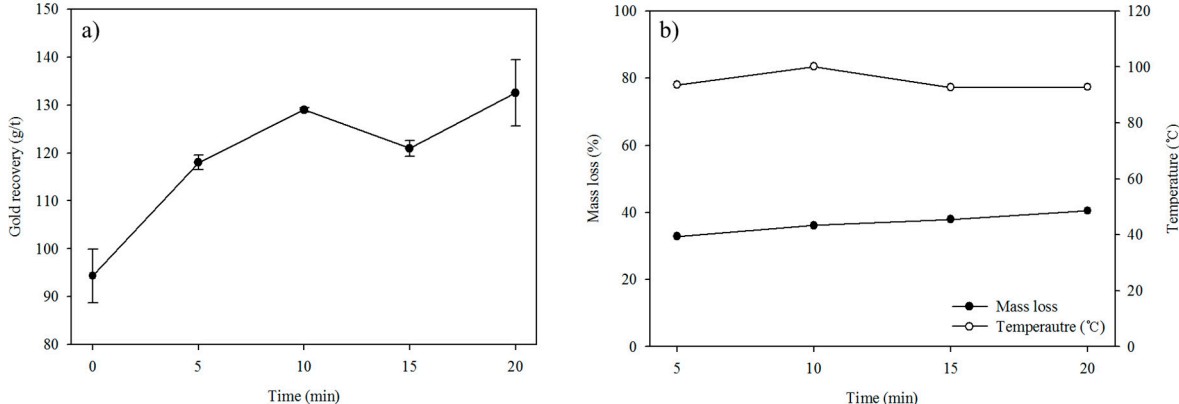

**Figure 14.** Effect of leaching time on (**a**) recovery of gold and (**b**) mass loss and reaction temperature from the refractory gold concentrate. The leaching conditions were an $HNO_3$ concentration of 2.0 M.

Leach residues were collected by performing microwave-assisted leaching experiments for each nitric acid concentration. When XRD analysis was conducted on these leach residues, quartz, pyrite, and sulfur were detected (Figure 15). While pyrite was detected with 3 M nitric acid, it disappeared with 5 M nitric acid. This means that the intensities of the pyrite peak decreased, while those of the gangue peak increased in the leach residues, thus indicating the dissolution of pyrite during microwave-assisted leaching. However, sulfur appeared in the leach residue at 3.0 M and 5.0 M nitric acid, even though it was not detected in the concentrate. This appears to be due to the reactions of the pyrite included in the refractory gold concentrate with nitric acid, as shown in reactions (3) and (4) [25].

$$FeS_2 + 8HNO_3 = Fe(NO_3)_3 + 2H_2SO_4 + 2H_2O + 5NO \tag{3}$$

$$2FeS_2 + 8HNO_3 = Fe_2(SO_4)_3 + S^0 + 8NO + 4H_2O \tag{4}$$

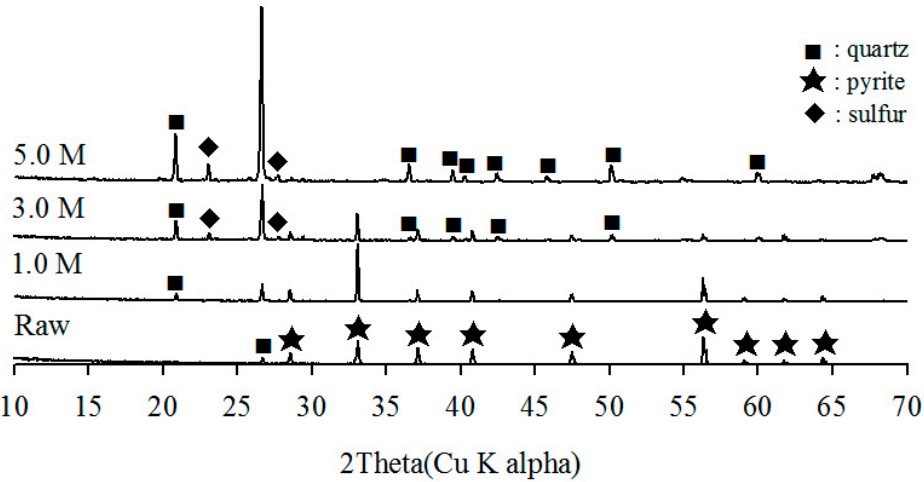

**Figure 15.** XRD patterns of raw and leach residue from nitric acid leaching at 1.0, 3.0 and 5.0 M. The leaching time was 15 min.

From the redox reactions above, it can be inferred that sulfur was generated from the leaching decomposition. However, most studies [13,19] have reported that $S^0$ is responsible for the blocking of the surface. The transformation of S species, such as $S^0$, on the surface is the most important intermediate during the leaching of sulfide minerals, and it is considered to be the main component hindering the dissolution of sulfide minerals. The results also revealed that a larger proportion of sulfur was transformed and the generated hydroxyl precipitated (Figure 16), which could be confirmed

by SEM-EDS. More extreme reaction conditions, such as the increase in nitric acid concentration from 1.0 M to 5.0 M, facilitated the decomposition of passivation species derived from metal ion dissolution and the liberation of gangue minerals from the sulfide surface. After leaching, SEM-EDS analysis was conducted on the leach residues (Figure 16). Based on the EDS analysis, the particles contain sulfur. Leaching increased with time, due to the oxidation of insoluble sulfides to soluble sulfate phases. A larger proportion of sulfur was transformed from the pyrite lattice in the refractory sulfide, thus leaving many vacant areas and microstructures, which can effectively liberate encapsulated gold and improve the recovery of gold.

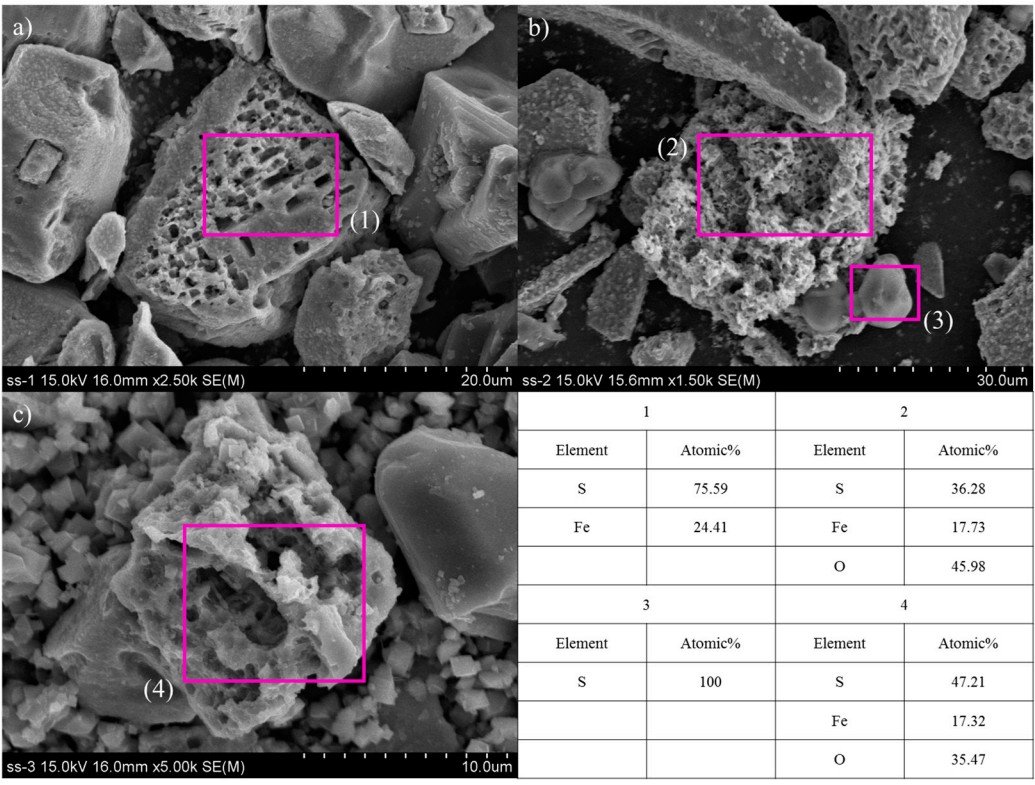

| 1 | | 2 | |
|---|---|---|---|
| Element | Atomic% | Element | Atomic% |
| S | 75.59 | S | 36.28 |
| Fe | 24.41 | Fe | 17.73 |
| | | O | 45.98 |
| 3 | | 4 | |
| Element | Atomic% | Element | Atomic% |
| S | 100 | S | 47.21 |
| | | Fe | 17.32 |
| | | O | 35.47 |

**Figure 16.** SEM images of the leach residues from nitric acid leaching at (a) 1.0 M, (b) 3.0 M and (c) 5.0 M. The leaching time was 15 min.

## 4. Conclusions

The leaching efficiencies of metal ions (As, Cu, Zn, Fe, and Pb) and the recovery of gold from the refractory gold concentrate were investigated using a microwave system. As the acid concentration increased, the metal ion leaching increased, but the leach residues for the low nitric acid concentration (> 0.5 M) consisted mainly of untreated pyrite. The S/Fe ratio determines the surface of the leach residue, which affects the efficiency of passive layer removal. S/Fe ratio in nitric acid leaching with a reaction time of 10 min reached 4.34 at 0.1 M and 4.20 at 0.5 M. It was found that a relatively lower proportion of iron was consumed, and the pyrite in the refractory gold concentrate did not significantly dissolve during nitric acid leaching at > 0.5 M.

In the refractory gold concentrate leaching experiments, nitric acid leaching at high-temperature could limit the decomposition of sulfide minerals, due to the passive layer in the refractory gold concentrate. During the leaching process, As, Cu, Fe, and Zn, leaching increased with time, due to the oxidation of insoluble sulfides to soluble sulfate phases. Initially, the As leaching rate was slow, but steadily increased with increased nitric acid concentration. For the refractory gold concentrate, the As, Cu, Fe, and Zn leaching increased from 7.28% to 84.0% (As), 20.8% to 85.8% (Cu), 9.19% to 98.7% (Fe), and 2.46% to 14.9% (Zn), upon increasing the $HNO_3$ concentration from 0.1M to 1.0M (80 °C,

S/L ration 5, leaching time 15 min). As the temperature increased from 80 to 120 °C, the Cu leaching rate constant increased from 0.18 to 0.36 min$^{-1}$ and the Fe leaching rate constant increased from 0.21 to 0.66 min$^{-1}$. However, Pb leaching decreased at > 80 °C, due to complex lead and passivation, by increased elemental sulfur formation from the high-temperature oxidation.

Microwave-assisted leaching experiments for gold recovery were conducted for the refractory gold concentrate. More extreme reaction conditions, such as the increase in nitric acid concentration from 1.0 to 5.0 M, facilitated the decomposition of passivation species derived from metal ion dissolution and the liberation of gangue minerals from the sulfide surface. From the comparison between the XRD patterns of the refractory gold concentrate and the leach residues after leaching with different nitric acid concentrations, it can be concluded that pyrite in the sulfide minerals can be destroyed. The SEM-EDS analyses of the leach residue showed that dissolution and decomposition of pyrite in the complex sulfide concentrate leave many vacant areas and microstructures, which can effectively liberate encapsulated gold and improve the recovery of gold.

The recovery rate of gold in the leach residue was improved with microwave-assisted leaching and the gold recovery was about 132.55 g/t after 20 min of the leaching experiment (nitric acid at 2.0 M), according to fire assays. The effect of the increase in nitric acid concentration was consistent with increased exposure to reactive sulfide minerals, which could effectively liberate, encapsulate and improve the gold recovery rate.

The current rapid decline in high-grade gold ores has made the mineral processing industry increasingly reliant on complex and refractory gold ores. At present, several approaches have been employed for mineral processing, including metallurgy and hydro-metallurgy. The leaching process is a method used in hydrometallurgy, which is used to leach base and precious metals from source materials. Particularly, percolation leaching methods, such as Heap leaching, dump leaching, bioleaching and in situ leaching, have been very effective in extracting metals from low grade ores, which could not otherwise be economically extracted. However, the challenge of this facility has been the handling of waste and control of environmental pollution caused by toxic leakages from heaps [26]. In addition, the formation of complex or refractory substances has limited its application. The main reason for this is that the complex or refractory ores require more processing and the various approaches used for gold recovery are challenging to perform in aqueous solutions, owing to surface passivation.

Gold-bearing ores contain pyrite, chalcopyrite, arsenopyrite, and galena, which interfere with gold recovery and lower its efficiency. Therefore, studying the effects of relational minerals on gold is important. Among the alternative processes, the use of microwave additives followed by leaching is important for the recovery of precious metals from complex or refractory minerals. To increase the recovery of gold, the microwave-assisted leaching process may be used as a process to address the formation of complex or refractory sulfide minerals.

**Supplementary Materials:** The following are available online at http://www.mdpi.com/2075-4701/10/5/571/s1, Table S1: The summary of leaching parameters for As, Cu, Zn, and Fe leaching using Equation (2). The leaching conditions were a reaction time of 15 min, reaction temperature of 80 °C, and HNO3 concentration of 0.1, 0.5, or 1.0 M. Table S2: The summary of leaching parameter for As, Cu, Zn, and Fe leaching, using Equation (2). The leaching conditions were a reaction temperature between 80 and 120 °C, HNO3 concentration of 1.0 M, and reaction time of 15 min.

**Author Contributions:** H.K.: Experiment, Investigation; E.M.: Formal analysis, Visualization; O.P.: Formal analysis; N.C.: Funding acquisition, Conceptualization; C.P.: Project administration, Conceptualization; K.C.: Writing-original draft, Writing-review & editing. All authors have read and agreed to the published version of the manuscript.

**Funding:** This study was supported by the Korea Ministry of Environment (MOE), South Korea, as an Advanced Industrial Technology Development Project (No. 2016000140010).

**Conflicts of Interest:** The authors declare no conflicts of interest.

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
