# Peer review of "Recovery of Gold from the Refractory Gold Concentrate Using Microwave Assisted Leaching"

_metals, doi:10.3390/met10050571_

Round 1
Reviewer 1 Report
Even though the authors presented their results clearly, I believe, the introduction needs to be improved to highlight the current status of this subject and the research problem. I have included my comments in the attached file.

Author Response
Dear Reviewer
Thank you for taking the time to review our paper.
The manuscript was revised in the aim to improve the overall quality of the writing in accordance with the recommendations of the reviewer. Thank you very much for your effort.
Point 1: Abstract and Introduction
Response 1: Your comment helped our manuscript substantially improved. We appreciate your precise evaluation. As you point out, we changed introduction. During the revision, Line 25-74.
Point 2: Discuss the significance of this work in industrial processes in addition to the summary of the results presented here.
Response 2: As you point out, we changed results. During the revision, Line 386-395.

Reviewer 2 Report
The manuscript proposed by the authors is interesting and well written. However, a few important points deserve more attention:
1) The results of nitric acid leaching tests should be compared with those of other studies in the literature on the use of nitric acid or other inorganic acids (sulfuric or hydrochloric);
2) The major part of the study (12 figures and 3 tables) presents results on the use of nitric acid alone, without the application of microwaves. In this context, it would be necessary to better justify the originality of these results compared to other comparable studies in the scientific literature.
3) In the results, there is a lack of control carried out under the same conditions but with and without the use of microwaves.
4) From a practical point of view, how will the acidic leachate containing a high concentration of nitric acid, high levels of iron and other metals be managed?
5) The practical feasibility of such a treatment approach should be discussed in relation to other pre-treatment techniques for refractory gold concentrates.
Author Response
Dear Reviewer
We greatly appreciate your thoughtful comments that helped improve the manuscript. We trust that all of your comments have been addressed accordingly in a revised manuscript. Thank you very much for your effort
Point 1: The results of nitric acid leaching tests should be compared with those of other studies in the literature on the use of nitric acid or other inorganic acids (sulfuric or hydrochloric);
Response 1: As you point out, we changed introduction. During the revision, Line 25-74.
Point 2: The major part of the study (12 figures and 3 tables) presents results on the use of nitric acid alone, without the application of microwaves. In this context, it would be necessary to better justify the originality of these results compared to other comparable studies in the scientific literature.
Point 3: In the results, there is a lack of control carried out under the same conditions but with and without the use of microwaves.
Response 2 and 3: For the leaching experiment of the refractory gold concentrate, a microwave was used. During the revision, Line 104.
In this study, we aim to recovery gold from invisible gold concentrate by investigating and metal ion leaching behaviour in nitric acid. In detail, gold is recovered by applying cyanide or non-cyanide solvent to the gold-bearing concentrate and dissolving the gold. However, this method can’t effectively dissolve gold because invisible gold is trapped by sulfide minerals as nanoparticle size or forms a solid solution. In addition, these solvents can dissolve not only the gold but also the sulfide minerals. Therefore, the consumption of cyanide or non-cyanide solvent increases, resulting in an increase in the gold-production cost. However, when a gold-bearing concentrate is subjected to microwave-nitric acid leaching, only the sulfide minerals are selectively and dissolved within a few minutes, and gold is not dissolved. In other words, when the sulfide minerals are dissolved and removed, the invisible gold trapped inside such minerals is separated. The invisible gold liberated from the sulfide minerals is left in insoluble residues while maintaining its own morphology and size. Of course, in this instance, silica minerals such as quartz and muscovite are also included in the insoluble residues without being dissolved by the nitric acid. However, the effect of the increase in nitric acid concentration was consistent with increased exposure to reactive sulfide minerals, which could effectively liberate encapsulated gold and improve its recovery rate.
Point 4: From a practical point of view, how will the acidic leachate containing a high concentration of nitric acid, high levels of iron and other metals be managed?
Response 4: I understand your concerns and agreed your comments.
Furthermore, the reaction rates of metal leaching also increase at nitric acid concentration. Further process for the recovery of metal from leaching solution is necessary such as solvent extraction, chemical precipitation and ion-exchange adsorption have been applied.
In particularly, a variety of ion-exchange resins, able to sequester metal ions from leaching solution, are commercially available. It is adsorption performance, such as the selectivity, can be easily adjusted. The focus of our further research will be the study of the recovery of metal from leaching solution using ion-exchange adsorption.
Point 5: The practical feasibility of such a treatment approach should be discussed in relation to other pre-treatment techniques for refractory gold concentrates.
Response 5: As you point out, we changed introduction. During the revision, Line 386-395.

Round 2
Reviewer 1 Report
Even though the authors improved the manuscript, I would like to suggest the following comments:
First 1-2 sentences of the abstract should be introduction to the subject – but you have started with the objective here
The newly added section in Conclusion section should be given in Introduction section as an introductory section to microwave pre-treatment. The reason is microwave pre-treatment has been justified as the correct solution to the research gap of this work.
This work gives some numbers relevant to the Gold production in China (https://www.sciencedirect.com/science/article/abs/pii/S0892687518302760), the largest gold producer in the world. Better to include recent gold production numbers in the initial part of the introduction, including the world production.
Conclusion – Include how we apply the results in this work in industrial gold leaching processes (i.e. your thoughts).
Author Response
Dear Reviewer
We greatly appreciate your thoughtful comments that helped improve the manuscript.
Point 1: First 1-2 sentences of the abstract should be introduction to the subject – but you have started with the objective here
Response 1: As you point out, we changed introduction. During the revision, Line: 10-14
Point 2: The newly added section in Conclusion section should be given in Introduction section as an introductory section to microwave pre-treatment. The reason is microwave pre-treatment has been justified as the correct solution to the research gap of this work. This work gives some numbers relevant to the Gold production in China (https://www.sciencedirect.com/science/article/abs/pii/S0892687518302760), the largest gold producer in the world. Better to include recent gold production numbers in the initial part of the introduction, including the world production. Conclusion – Include how we apply the results in this work in industrial gold leaching processes (i.e. your thoughts).
Response 2: Thanks for the reviewer's comment. During revision line: 385-402
Reviewer 2 Report
In my opinion, the corrections and responses presented by the authors are adequate.Author Response
Dear Reviewer
Thank you for taking the time to review our paper.
Round 3
Reviewer 1 Report
I believe, the newly added section in the conclusion section is based on the suggested reference and better to include it there.
Author Response
Thanks for the reviewer's comment. Reference 27 add.